ecology/biochemistry/analytical chemistry

skin, bone, archaeology, cross-disciplinary sciences, analytical chemistry, stable isotope analysis

**Author for correspondence:**
Sean P. Doherty
e-mail: sean@palaeome.org

# A modern baseline for the paired isotopic analysis of skin and bone in terrestrial mammals

Sean P. Doherty[1,2], Matthew J. Collins[3,4],
Alison J. T. Harris[5], Ainara Sistiaga[4,6], Jason Newton[7]
and Michelle M. Alexander[2]

[1]Department of Archaeology, University of Exeter, Exeter EX4 4QE, UK
[2]BioArCh, Department of Archaeology, University of York, York YO10 5DD, UK
[3]McDonald Institute for Archaeological Research, University of Cambridge, Cambridge CB2 3ER, UK
[4]Section for Evolutionary Genomics, Globe Institute, University of Copenhagen, Copenhagen 1353, Denmark
[5]Department of Archaeology, Memorial University of Newfoundland, St John's A1C 5S7, Canada
[6]Earth, Atmospheric and Planetary Sciences Department, Massachusetts Institute of Technology, Cambridge, MA 02139, USA
[7]NERC Life Sciences Mass Spectrometry Facility, Scottish Universities Environmental Research Centre, East Kilbride G75 0QF, UK

SPD, 0000-0002-5503-2734; MJC, 0000-0003-4226-5501;
AJTH, 0000-0001-8017-7188; AS, 0000-0003-1732-7010;
JN, 0000-0001-7594-3693; MMA, 0000-0001-8000-3639

We present the isotopic discrimination between paired skin and bone collagen from animals of known life history, providing a modern baseline for the interpretation of archaeological isotopic data. At present, the interpretation of inter-tissue variation ($\Delta_{(skin-bone)}$) in mummified remains is based on comparisons with other archaeological material, which have attributed divergence to their contrasting turnover rates, with rapidly remodelling skin collagen incorporating alterations in environmental, cultural and physiological conditions in the months prior to death. While plausible, the lack of baseline data from individuals with known life histories has hindered evaluation of the explanations presented. Our analysis of a range of animals raised under a variety of management practices showed a population-wide trend for skin collagen to be depleted in $^{13}C$ by −0.7‰ and enriched in $^{15}N$ by +1.0‰ relative to bone collagen, even in stillborn animals. These results are intriguing and difficult to explain using current knowledge; however, on the basis of the findings reported here, we

caution any results which interpret simply on differing turnover rates. We hypothesize that there may be a consistent difference in the routing of dietary protein and lipids between skin and bone, with potentially on-site synthesis of non-essential amino acids using carbon and nitrogen that have been sourced via different biochemical pathways.

# 1. Introduction

With the $\delta^{13}$C and $\delta^{15}$N of proteinaceous tissue reflecting an individual's dietary intake and physiological condition during development [1–5], the paired analysis of skin collagen—remodelling on the scale of weeks and months [6,7]—and bone collagen—on the order of years and decades [8,9]—has been used to examine short- and long-term trends in diet, migration, health and cultural practices [10–19]. While variations in amino acid composition typically result in different tissues within the same individual displaying divergent isotopic values [20–22], bone and skin collagen are both dominated by type 1 collagen and possess highly similar amino acid profiles [23–25]. Inter-tissue isotopic variation ($\Delta_{(skin–bone)}$) is therefore frequently attributed to their contrasting temporal resolutions.

However, with isotope ecology studies rarely focusing on the analysis of either of these tissues, much of our understanding comes from archaeological studies where skin has survived through spontaneous and anthropogenic mummification. Here, $\Delta_{(skin–bone)}$ discrimination has been attributed to alterations in cultural and environmental conditions in the months prior to death, including substantial dietary change and/or geographical movement, as well as physiological and nutritional stress [11–19]. While these interpretations are plausible, the lack of baseline data from individuals with known life histories confounds evaluation of the explanations presented. $\Delta_{(skin–bone)}$ discrimination in archaeological populations may be additionally influenced by differential decomposition patterns [26] and the presence of exogenous contaminants [27–29].

While considerable effort has been made to optimize the extraction and analysis of collagen from archaeological bone (i.e. [30–33]), only recently protocols for the isotopic analysis of archaeological skin have been reviewed [28,34,35]. These highlight the necessity for sample pretreatment including lipid extraction and filtration of the collagen substrate to produce samples with acceptable collagen quality indicators (collagen yield, %C, %N and C:N ratio). Many archaeological skin samples fail to meet these quality indicators due to inadequate pretreatment, often producing elevated C:N ratios (greater than 3.5) indicative of the presence of carbon-rich contaminants, such as lipids, which may artificially reduce $\delta^{13}$C values due to the carbon fractionation that occurs during lipid synthesis [36].

Consequently, the only modern mammalian skin sample known to have undergone pretreatment (solvent extraction of lipids; demineralization in hydrochloric acid (HCl); gelatinization of the acid-insoluble fraction in pH 3 water; filtration and freeze-drying) and produced an acceptable C:N ratio (3.3) comes from a single pig [28]. To test the null hypothesis that there is no significant difference between $\delta^{13}$C and $\delta^{15}$N values in skin and bone collagen within an individual, we report inter-tissue discrimination in 25 sheep (*Ovis aries*) and five pigs (*Sus scrofa*) with known life histories, examining the impact of sex, age and health status.

# 2. Materials and methods

## 2.1. Sample selection

Paired skin and bone samples were obtained from 25 sheep raised across three flocks in the UK. Where available, husbandry information including sex, age, date of birth, date of death and health status was obtained from the farmer.

Flock 1 ($n = 14$) were raised in East Yorkshire, on a diet of local grazing supplemented by hay and protein concentrates. The animals were sent for slaughter at various ages between November 2014 and December 2015. Tissue samples were collected from two late-term stillborn lambs (SH10, SH11) and lambs that had died at 2 (SH09) and 24 (SH13) days old from natural causes. Two sheep had been in poor health shortly before death. SH03—six-month-old Shetland ram—had been treated for pneumonia in the months prior to death, which had reduced the animals' appetite, weight gain and growth. SH12—5-year-old Shetland ewe—was humanely euthanized due to flystrike, a condition where parasitic flies lay eggs in the wool and the resulting maggots bury into the skin feeding off the

flesh. This had severely weakened and debilitated the ewe, resulting in cessation of wool production and fleece shedding (electronic supplementary material, figure S1).

Flock 2 ($n = 2$) were raised in Leicestershire on a diet of local grazing supplemented with protein concentrates. SH28 was crushed by the mother at one month old, and SH29 was slaughtered at 20 months old. Samples categorized as Flock 3 ($n = 10$) came from an abattoir in Nottinghamshire, had been raised locally and were all aged between eight and nine months. Further husbandry information was unavailable. Paired bone and skin samples from five pigs ($n = 5$) were obtained from an abattoir in Nottinghamshire, for which no husbandry information was unavailable.

## 2.2. Sample preparation

Sample preparation was conducted at the University of York using analytical grade or above reagents, and all water was deionized and distilled.

### 2.2.1. Skin collagen

A methodology was adopted in line with published analyses of modern and archaeological skins for stable isotope analysis and radiocarbon dating which had produced acceptable collagen quality indicators [35]. Shortly after death, the animals were flayed, and their skins stored at −5°C prior to analysis. In an effort to remove other proteinaceous material, samples were taken from the follicle sparse belly region, and all visible signs of hair and muscle were removed. The samples were subsequently freeze dried for 48 h and ground to a powder using a ball mill (Retsch MM400). The lipids were extracted via solvent extraction dichloromethane/methanol (2 : 1 v/v) by ultrasonication for 1 h, with the supernatant removed and solvent replaced every 15 min. The skin was briefly demineralized in 0.6 M HCl at 4°C for 1 h, rinsed with distilled water and gelatinized in pH 3 0.001 M HCl for 48 h. The supernatant containing the soluble collagen was filtered (60–90 µm Ezee-Filter, Elkay Laboratories, UK), frozen and freeze dried.

### 2.2.2. Bone collagen

The left metacarpal in sheep, and left humerus in pigs, was removed from the carcass and stored at −20°C prior to analysis. Adhering tissues were removed, and approximately 1 g of bone sampled from the anterior mid-shaft. The bone was cleaned in distilled water, freeze dried for 48 h to remove excess moisture and then ground to a powder. The lipids were extracted as with skin via ultrasonication for 2 h, and the bone was subsequently demineralized in 0.6 M HCl at 4°C for 4 days, gelatinized, filtered, frozen and freeze dried as above. Bone and skin samples (0.8–1.1 mg) were weighed out in duplicate into 4 × 3.2 mm tin capsules (Elemental Microanalysis, Okehampton, UK) for analysis.

## 2.3. Stable isotope analysis

Isotope ratio determinations were carried out at the Natural Environment Research Council Life Sciences Mass Spectrometry Facility, East Kilbride, on a ThermoElectron DeltaPlusXP (Thermo Fisher Scientific, Bremen, Germany) with an Elementar Pyrocube elemental analyser (Elementar UK Ltd). Sample data were reported in standard delta per mil notation ($\delta$‰) relative to Vienna Pee Dee Belemnite (V-PDB) ($\delta^{13}C$) and atmospheric air (AIR) ($\delta^{15}N$) international standards. Three laboratory reference materials were interspersed within the measurement run to correct for linearity and instrument drift. Each of the laboratory reference materials is checked regularly against international standards USGS40 and USGS41. Following the calculations outlined in Szpak et al. [37], the total analytical uncertainty was estimated to be ± 0.19‰ for $\delta^{13}C$ and ± 0.20‰ for $\delta^{15}N$.

## 2.4. Statistical analysis

Statistical testing was carried out using the IBM SPSS Statistics 26 software package. Shapiro–Wilks test for normality indicated $\delta^{13}C$ and $\delta^{15}N$ data did not conform to a normal distribution ($p < 0.01$); thus, the resultant statistical tests were non-parametric in nature. Significance of differences between bone and skin values determined using a Wilcoxon signed-rank test for paired samples. Significance of differences in $\Delta_{(skin–bone)}$ values between species, sexes and health status groups was determined using

an independent $t$-test. The correlation between age and $\Delta_{(skin-bone)}$ was determined through the Pearson correlation coefficient.

# 3. Results and discussion

## 3.1. Collagen quality indicators

Average collagen yields (dry mass) of 67% and 20% were obtained from skin and bone, respectively. This is well above the 2–4% threshold used to identify problematic samples [38,39]. Skin %C ranges from 41.4% to 46.1%, and %N values from 15.2% to 16.2% (table 1). Bone %C ranged from 39.6% to 47.6%, and %N from 14.9% to 16.9%, consistent with those reported in modern type I collagen-dominated tissues [38].

Atomic C:N ratios in skin ranged from 3.10 to 3.26 (mean 3.24), and in bone from 3.14 to 3.31 (mean 3.19). Skin collagen (type I) has a theoretical C:N ratio of 3.11 [35], though in practice, integrity is often measured against the 2.9–3.5 range used for bone collagen [40]. The disparity between observed and theoretical C:N ratios in skin may indicate the presence of non-collagenous proteins such as elastin (C:N 5.8) and epithelial keratins (C:N 3.4) which account for 5% of the protein fraction [41]. Like collagen, both are glycine rich [42–44] and therefore unlikely to impact the overall isotope signal in bulk analyses.

## 3.2. Isotopic discrimination between paired samples

Across all individuals, skin was on average depleted in $^{13}$C by –0.7‰ (s.d. 0.5‰) and enriched in $^{15}$N by +1.1‰ (s.d. 0.7‰) relative to bone (figure 1). The mean differences in both carbon ($Z = -4.70$, $p \le 0.001$) and nitrogen ($Z = -4.54$, $p \le 0.001$) isotopic values were significant (table 2). Within sheep, skin was depleted in $^{13}$C by –0.8‰ (s.d. 0.5‰) and enriched in $^{15}$N by +1.1‰ (s.d. 0.4‰) relative to bone, of which both mean differences are significant (carbon: $Z = \le 4.37$, $p \le 0.001$; nitrogen: $Z = -4.54$, $p \le 0.001$). The isotopic discrimination between tissues was lower in pigs, with skin on average depleted in $^{13}$C by –0.3‰ (s.d. 0.4‰) and enriched in $^{15}$N by +0.8‰ (s.d. 0.2‰), of which only the mean difference in nitrogen values is significant (carbon: $Z = -1.48$, $p = 0.14$; nitrogen: $Z = -2.02$, $p = 0.04$).

Mean $\Delta^{13}$C$_{(skin-bone)}$ values were comparable with those observed in other modern animals (figure 2) and broadly mid-range of those in archaeological populations (–2 to +1‰). Mean $\Delta^{15}$N$_{(skin-bone)}$ values in modern animals are lower than those observed in archaeological populations. It has been speculated that the greater susceptibility of skin to chemical and structural modification in the burial environment may be responsible for the elevated skin $\delta^{15}$N values observed in mummified individuals [12,15,19], as peptide bond hydrolysis preferentially eliminates isotopically lighter nitrogenous compounds resulting in an enrichment in the remaining protein [35,45]. While this has the potential to influence the nitrogen isotope values, it is clear from modern animals that an enrichment of at least 2.9‰ in the skin, relative to bone, can occur even in the absence of any diagenetic explanation.

## 3.3. Species

Mean $\Delta^{13}$C$_{(skin-bone)}$ values in sheep are marginally more depleted than those in pigs ($t_{28} = 2.1$, $p = 0.042$) although not for $\Delta^{15}$N$_{(skin-bone)}$ values ($t_{28} = -0.85$, $p = 0.41$). This difference may be due in part to the imbalanced sample sizes, along with differences in their management. Pigs—of which the majority in the UK are reared indoors—often display less isotopic variation than free-ranging ruminants [46–48] as plant isotope values can vary considerably within a small area due to variable mycorrhizal activity and areas of localized defecation [49]. The sheep from Flock 1 were all raised over the same five acres but exhibit a range of 2.8‰ and 2.5‰ in skin and bone $\delta^{13}$C values, and 1.7‰ and 2.1‰ in skin and bone $\delta^{15}$N values, respectively; isotopic variation comparable with that observed by von Holstein *et al.* [22] in wool keratin and bone collagen from a single flock.

## 3.4. Sex and age

As with archaeological populations [16,18,19], the magnitude of the isotopic discrimination between skin and bone was not related to the animals' sex (carbon: $t_{12} = 0.20$, $p = 0.84$; nitrogen: $t_{12} = 0.96$, $p = 0.35$).

**Table 1.** Husbandry information, isotopic values ($\delta^{13}C$, $\delta^{15}N$) and elemental composition of skin and bone samples. Sex: M = male, F = female; health: G = good, P = poor, DFM = died during first month of life; — = no data.

| ID | species | age | sex | health | skin | | | | | bone | | | | | $\Delta_{(skin-bone)}$ | |
|---|---|---|---|---|---|---|---|---|---|---|---|---|---|---|---|---|
| | | | | | %C | %N | C/N | $\delta^{13}C$ (‰) | $\delta^{15}N$ (‰) | %C | %N | C/N | $\delta^{13}C$ (‰) | $\delta^{15}N$ (‰) | $\Delta^{13}C$ (‰) | $\Delta^{15}N$ (‰) |
| SH02 | sheep | 6 months | M | G | 45.5 | 16.0 | 3.3 | −25.8 | 9.2 | 43.8 | 16.1 | 3.2 | −24.7 | 8.2 | −1.1 | +1.0 |
| SH03 | sheep | 6 months | M | P | 42.5 | 16.2 | 3.3 | −26.0 | 10.0 | 39.6 | 15.7 | 3.2 | −24.2 | 7.9 | −1.7 | +2.1 |
| SH04 | sheep | 11 months | M | G | 43.8 | 16.1 | 3.2 | −25.8 | 8.4 | 41.0 | 15.2 | 3.2 | −25.0 | 7.9 | −0.8 | +0.6 |
| SH05 | sheep | 10 months | M | G | 41.9 | 16.0 | 3.1 | −25.7 | 8.4 | 42.8 | 15.8 | 3.2 | −24.6 | 7.4 | −1.1 | +1.0 |
| SH06 | sheep | 10 months | M | G | 43.2 | 16.2 | 3.1 | −26.0 | 8.9 | 40.2 | 14.9 | 3.2 | −24.6 | 7.2 | −1.4 | +1.7 |
| SH07 | sheep | 10 months | F | G | 42.9 | 16.1 | 3.1 | −25.8 | 9.0 | 41.9 | 15.6 | 3.2 | −24.7 | 8.4 | −1.2 | +0.6 |
| SH08 | sheep | 13 months | F | G | 42.9 | 15.8 | 3.1 | −26.0 | 9.7 | 43.5 | 16.2 | 3.1 | −24.8 | 8.5 | −1.2 | +1.2 |
| SH09 | sheep | 2 days | M | DFM | 42.5 | 16.1 | 3.1 | −24.7 | 8.4 | 44.9 | 16.5 | 3.2 | −23.9 | 7.7 | −0.8 | +0.7 |
| SH10 | sheep | stillborn | F | DFM | 43.6 | 15.9 | 3.2 | −25.1 | 8.7 | 43.8 | 16.0 | 3.2 | −24.4 | 7.7 | −0.7 | +1.0 |
| SH11 | sheep | stillborn | M | DFM | 43.5 | 16.0 | 3.2 | −24.9 | 8.1 | 44.9 | 16.6 | 3.2 | −23.9 | 7.4 | −1.0 | +0.7 |
| SH12 | sheep | 5 years | F | P | 43.7 | 15.2 | 3.3 | −26.0 | 9.3 | 42.1 | 15.6 | 3.2 | −24.6 | 6.4 | −1.5 | +2.9 |
| SH13 | sheep | 24 days | M | DFM | 43.2 | 16.1 | 3.3 | −23.3 | 7.6 | 44.2 | 16.0 | 3.2 | −23.3 | 7.0 | 0.0 | +0.6 |
| SH16 | sheep | 18 months | M | G | 44.2 | 15.9 | 3.2 | −25.4 | 8.0 | 45.7 | 16.9 | 3.2 | −24.5 | 6.9 | −0.9 | +1.1 |
| SH28 | sheep | 20 months | F | G | 44.7 | 15.6 | 3.4 | −25.3 | 7.3 | 45.7 | 16.8 | 3.2 | −25.2 | 5.8 | −0.1 | +1.5 |
| SH29 | sheep | 1 month | F | DFM | 43.9 | 15.3 | 3.3 | −24.7 | 8.0 | 45.1 | 16.6 | 3.2 | −24.0 | 7.1 | −0.6 | +0.9 |
| SH30 | sheep | 8–9 months | — | G | 43.6 | 15.6 | 3.3 | −25.6 | 9.7 | 44.0 | 16.1 | 3.2 | −25.3 | 8.5 | −0.2 | +1.2 |
| SH31 | sheep | 8–9 months | — | G | 41.4 | 15.4 | 3.3 | −25.5 | 9.5 | 43.6 | 16.7 | 3.2 | −25.1 | 8.0 | −0.4 | +1.4 |
| SH32 | sheep | 8–9 months | — | G | 44.3 | 15.8 | 3.1 | −25.4 | 10.1 | 44.1 | 16.8 | 3.2 | −25.0 | 9.1 | −0.5 | +1.0 |
| SH33 | sheep | 8–9 months | — | G | 43.8 | 15.5 | 3.3 | −25.6 | 10.0 | 43.5 | 16.5 | 3.2 | −25.3 | 9.0 | −0.3 | +1.1 |
| SH34 | sheep | 8–9 months | — | G | 43.5 | 15.6 | 3.3 | −25.8 | 10.0 | 43.1 | 15.8 | 3.2 | −25.3 | 8.8 | −0.4 | +1.2 |
| SH35 | sheep | 8–9 months | — | G | 43.8 | 15.5 | 3.2 | −26.0 | 10.5 | 45.0 | 16.4 | 3.2 | −25.5 | 9.4 | −0.5 | +1.1 |

(*Continued.*)

**Table 1.** (*Continued.*)

| ID | species | age | sex | health | skin | | | | | bone | | | | | Δ(skin–bone) | |
|---|---|---|---|---|---|---|---|---|---|---|---|---|---|---|---|---|
| | | | | | %C | %N | C/N | $\delta^{13}$C (‰) | $\delta^{15}$N (‰) | %C | %N | C/N | $\delta^{13}$C (‰) | $\delta^{15}$N (‰) | $\Delta^{13}$C (‰) | $\Delta^{15}$N (‰) |
| SH36 | sheep | 8–9 months | — | G | 43.6 | 15.7 | 3.3 | −25.4 | 7.4 | 44.2 | 15.7 | 3.2 | −24.8 | 7.7 | −0.7 | −0.3 |
| SH37 | sheep | 8–9 months | — | G | 43.1 | 15.3 | 3.2 | −25.9 | 7.2 | 44.6 | 16.4 | 3.2 | −24.8 | 8.2 | −1.2 | −0.9 |
| SH38 | sheep | 8–9 months | — | G | 46.1 | 15.7 | 3.3 | −26.4 | 9.1 | 45.1 | 15.8 | 3.2 | −24.9 | 6.6 | −1.4 | +2.5 |
| SH39 | sheep | 8–9 months | — | G | 43.6 | 15.5 | 3.4 | −25.8 | 9.6 | 45.1 | 15.5 | 3.2 | −25.4 | 8.0 | −0.4 | +1.6 |
| PG01 | pig | — | — | — | 43.6 | 15.8 | 3.2 | −23.3 | 4.1 | 44.3 | 16.7 | 3.2 | −23.5 | 3.5 | +0.2 | +0.6 |
| PG02 | pig | — | — | — | 43.9 | 15.9 | 3.2 | −23.3 | 4.3 | 43.9 | 16.6 | 3.1 | −23.3 | 3.5 | −0.1 | +0.7 |
| PG03 | pig | — | — | — | 42.7 | 15.7 | 3.2 | −23.0 | 5.6 | 46.4 | 15.9 | 3.2 | −22.6 | 5.0 | −0.4 | +0.6 |
| PG04 | pig | — | — | — | 43.7 | 15.8 | 3.2 | −23.0 | 5.8 | 47.6 | 15.8 | 3.2 | −22.3 | 4.6 | −0.7 | +1.1 |
| PG05 | pig | — | — | — | 43.4 | 15.8 | 3.2 | −23.6 | 4.5 | 44.2 | 16.2 | 3.2 | −23.0 | 3.5 | −0.6 | +1.0 |

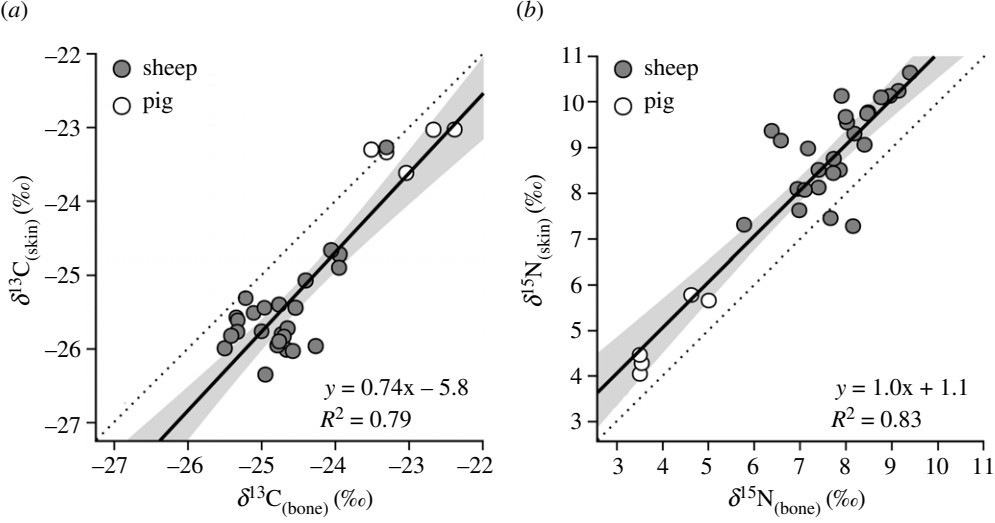

**Figure 1.** Isotope values in modern skin and bone samples. Comparison of (a) $\delta^{13}$C and (b) $\delta^{15}$N values in skin collagen and bone collagen pairings. Solid line = linear trend line with 95% CI (grey shading); dashed line = 1:1.

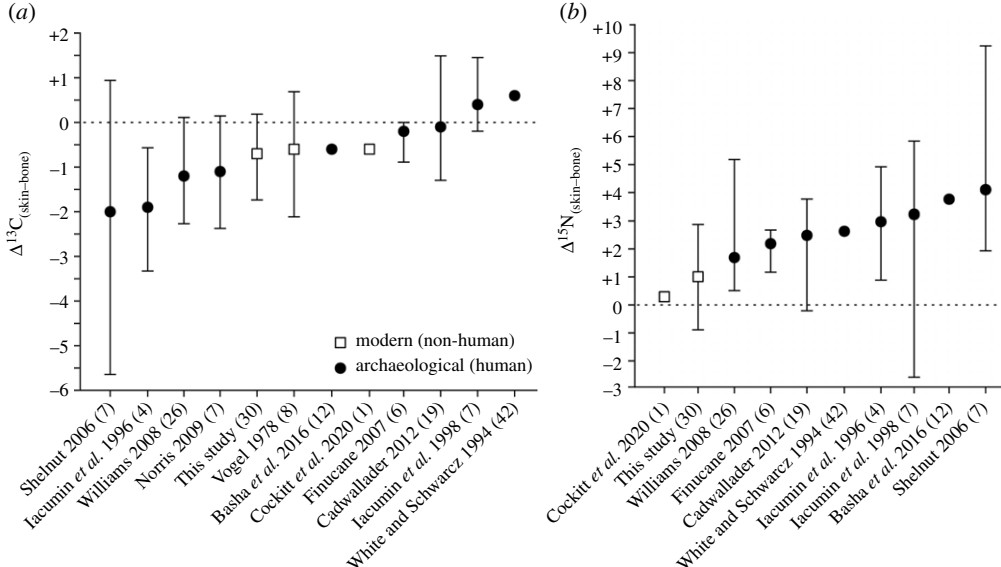

**Figure 2.** Mean and range of $\Delta^{13}$C$_{(skin-bone)}$ and $\Delta^{15}$N$_{(skin-bone)}$ values reported in modern and archaeological studies. Number of paired samples in parentheses.

**Table 2.** Isotopic discrimination between skin and bone in modern sheep and pigs.

| species | $n$ | mean (‰) | min (‰) | max (‰) | s.d. |
|---|---|---|---|---|---|
| | | | $\Delta^{13}$C$_{(skin-bone)}$ | | |
| all | 30 | −0.7 | −1.72 | +0.17 | 0.5 |
| sheep | 25 | −0.7 | −1.72 | −0.01 | 0.5 |
| pig | 5 | −0.3 | −0.71 | +0.17 | 0.4 |
| | | | $\Delta^{15}$N$_{(skin-bone)}$ | | |
| all | 30 | +1.1 | +0.92 | +2.90 | 0.7 |
| sheep | 25 | +1.1 | +0.92 | +2.90 | 0.8 |
| pig | 5 | +0.8 | +0.55 | +1.13 | 0.2 |

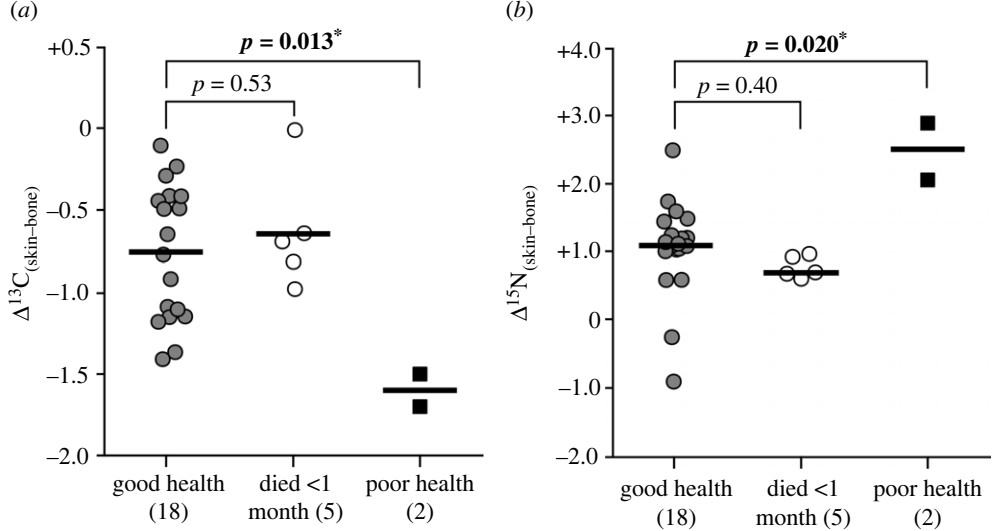

**Figure 3.** (a) $\Delta^{13}C_{(skin-bone)}$ and (b) $\Delta^{15}N_{(skin-bone)}$ isotopic discrimination in sheep by health status. Horizontal line = group mean, number of samples in parentheses. Significance of difference between groups determined using an independent samples $t$-test, equal variances assumed based on Levene's test. *Statistically significant differences ($p \leq 0.05$).

A significant positive correlation was observed between $\Delta^{15}N_{(skin-bone)}$ values and age ($r = 0.50$, $n = 25$, $p = 0.010$); however, this was heavily influenced by the oldest individual (SH12, 5 years old, $\Delta^{15}N_{(skin-bone)} = +2.9‰$), which had been in poor health prior to death (see §3.5). No correlation was found between $\Delta^{13}C_{(skin-bone)}$ values and age ($r = -0.27$, $n = 25$, $p = 0.21$).

The skins of two stillborn lambs (SH10, SH11) and a 2-day-old lamb (SH09) all showed a depletion in $^{13}C$ (mean: −0.8‰) and enrichment in $^{15}N$ (mean: +0.8‰) consistent with the wider population. They had received all of their nutrition from their mother *in utero* with the same nutritional pool supporting the development of both tissues [50]. This suggests that the variation between these tissues is present regardless of turnover rate and may be the consequence of the unequal allocation of isotopically distinct amino acids (AA) during tissue synthesis.

## 3.5. Health status

The sheep known to have been in poor health before death—SH03 (suffered from pneumonia) and SH12 (euthanized following flystrike)—had a mean depletion of −1.6‰ in skin $^{13}C$ composition relative to bone (s.d. 0.2‰) and a mean enrichment of +2.5‰ in $^{15}N$ (s.d. 0.6‰) (figure 3). The mean differences in both carbon and nitrogen $\Delta_{(skin-bone)}$ values between these adults in poor health and adults in good health are significant (carbon: $t_{18} = -2.75$, $p = 0.013$; nitrogen: $t_{2.6} = 2.71$, $p = 0.020$). Values in lambs that had died less than one month old (including stillborns) were not significantly different from those in good health (carbon: $t_{21} = -0.64$, $p = 0.53$; nitrogen: $t_{21} = -0.85$, $p = 0.40$).

Physiological stress resulting from acute illness or malnutrition can alter the isotopic value of tissues [2,4,51–56]. When fatal, these are often most pronounced in tissue laid down in the final 7–14 days of life [4]. While most studies have focused on inert tissues, the rapid turnover of skin collagen may enable the recording of short-term stresses which are not visible in bone. Starvation typically results in an elevation in $\delta^{15}N$ values as the body enters a negative nitrogen balance, with the catabolism of the body's own protein resulting in a fractionation typical of trophic level increases [2,4,51,52]. The impact of physiological stress on $\delta^{13}C$ values is less defined [57], though a number of studies have observed a depletion in response to starvation, potentially due to nutritional ketosis and the utilization of $^{13}C$-depleted fat deposits as an energy source in the absence of dietary carbohydrate [2,4,54,55].

The catabolism of lipid reserves and proteins is a plausible explanation for the values observed in these sheep. Their poor health is known to have led to a reduction in weight, and, in the case of SH12, cessation of wool production and fleece shedding which only occurs during severe nutritional deficiency [58]. Although a very small sample size, it indicates that physiological stress in the final months of life is a potential explanation for the $^{13}C$-depleted and $^{15}N$-enriched values seen in the skins of many archaeological populations.

# 4. Conclusion

Contrary to the assumptions made by previous studies, we reject the hypothesis that $\delta^{13}$C and $\delta^{15}$N variation between skin and bone collagen in an individual is the result of their contrasting turnover rates. In this analysis, a variety of animals raised under different management practices, slaughtered in different years and at different ages, showed a population-wide trend for skin to have a lower $\delta^{13}$C and higher $\delta^{15}$N than bone. If the explanation was simply turnover rate, we would expect an average near zero, but with considerable variation. It is also unlikely to reflect protein or amino acid composition due to the minor variation between these tissues.

We instead postulate a consistent difference in the routing of dietary protein and lipids between skin and bone, with potentially on-site synthesis of non-essential amino acids using carbon and nitrogen that have been sourced via different biochemical pathways [59,60]. In the absence of secondary influences (i.e. dietary change, physiological stress or diagenesis), this results in skin having a lower $\delta^{13}$C and higher $\delta^{15}$N than bone. While we are unable to adequately explain this result currently, future isotope analysis of individual AAs in paired samples may provide valuable insight.

This study provides an important dataset for the interpretation of skin–bone isotopic discrimination in archaeological studies, reporting the range of values that can be expected in the absence of geographical or dietary change and diagenesis. The elevated $\Delta^{15}$N$_{(skin–bone)}$ and lower $\Delta^{13}$C$_{(skin–bone)}$ values observed in animals known to be in poor health highlight the significant potential of the isotopic analysis of skin for the study of health and palaeopathology. Similar trends have been observed in numerous mummified individuals and may indicate a period of acute physiological stress prior to death. $\delta^{15}$N and $\delta^{13}$C isotope analysis of individual AAs in skin samples may in future offer a route to exploring metabolic shifts associated with poor health, and in the case of processed skin such as parchment, offer insights into animal management and skin selection.

Ethics. No animals were raised or slaughtered for this study. Skin and bone samples were obtained as by-products of animals that had entered the food chain for human consumption. The authors were not involved in the selection or timing of the animals' slaughter, with all husbandry decisions made by the animals' registered owners. Samples were either obtained directly from UK Food Standards Agency (FSA) licensed slaughterhouses or from the animals' owner after the carcass had returned from an FSA licensed slaughterhouse.

Data accessibility. All data generated in this study are presented in the article and electronic supplementary material file.

Authors' contributions. S.P.D.: conceptualization, data curation, formal analysis, funding acquisition, investigation, methodology, project administration, resources, visualization, writing—original draft, writing—review and editing; M.J.C.: conceptualization, formal analysis, funding acquisition, investigation, methodology, resources, supervision, writing—review and editing; A.J.T.H.: investigation, writing—review and editing; A.S.: investigation, writing— review and editing; J.N.: formal analysis, investigation, writing—review and editing; M.M.A.: conceptualization, formal analysis, investigation, methodology, writing—review and editing. All authors gave final approval for publication and agreed to be held accountable for the work performed therein.

Competing interests. At the time of writing, Professor Matthew Collins and Dr Michelle Alexander are Board Members of Royal Society Open Science but were not involved in the review or assessment of the paper.

Funding. This research was supported by NERC Life Sciences Mass Spectrometry Facility grant no. EK259–14/15. S.P.D. was supported by the AHRC White Rose College of Arts and Humanities Doctoral Training Partnership (Award Ref. 1489527). M.J.C is supported by the Danish National Research Foundation-DNRF128.

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
