## [Peer Review File · Royal Society Open Science]

Review History

RSOS-210316.R0 (Original submission)

Review form: Reviewer 1

Is the manuscript scientifically sound in its present form?

Yes

Are the interpretations and conclusions justified by the results?

No

Is the language acceptable?

Yes

Do you have any ethical concerns with this paper?

No

Have you any concerns about statistical analyses in this paper?

No

Recommendation?

Major revision is needed (please make suggestions in comments)

Comments to the Author(s)

Although I have selected "Major revision", I want to be clear that overall, I think this is a high-quality paper that will make a worthwhile contribution.

I base my response only on the fact that, unless I am misinterpreting the data, I think that the authors could dig deeper into their data and provide a more nuanced explanation for the trends that they observe, and perhaps even state that their findings appear to challenge some of the fundamental principles that guided their experimental design. In the event that I am wrong, my input should at least provide the authors guidance for where to clarify their text. Details are below.

Line 82...this may be a good place to describe that lipid contamination, generally indicated by high CN ratios, often results in reduced $\delta^{13}\text{C}$ values because of the carbon fractionation that occurs during lipid synthesis as described by DeNiro and Epstein.

88...prior to these passages, the authors have only described the processes of lipid extraction and filtration (lines 65-66). Here, the authors need to define or describe gelatinization, freeze-drying, and demineralization.

98...see previous comment about line 88.

175...is this yield based on dry mass?

178...I believe this should be in past tense?

198...change to "...the elevated skin $\delta^{15}\text{N}$ values observed..."

202...rephrase this to be more precise. It is clear from modern animals that an enrichment of at least 1.1 per mil in the skin, relative to bone, can occur even in the absence of any diagenetic explanation.

231...Lübcker et al (2020. Conservation Physiology DOI: 10.1093/conphys/coaa060) conducted a relevant analysis, and offer insight into the isotope dynamics of specific amino acids during allocation from mothers to fetal tissue. Consider including their work into this discussion.

246...the seasonal variation in the $\delta^{13}\text{C}$ of bone is small, with medians ranging only from -24 to -25. However, the seasonal variation in the $\delta^{15}\text{N}$ of bone is surprisingly large, with medians from 6 to 8. Given that $\delta^{15}\text{N}$ increases by roughly 3 per mil with a trophic level (an approximation, but a useful one), this change in bone represents nearly a trophic level. This is unexpected, because as described in the manuscript, bone should be a long-term average. However, nearly all of the sheep were less than one year of age; perhaps they had not yet lived through enough seasons for the bone to become more homogenous over time. Or perhaps in the first year of life, bone is still more of a reservoir for stored energy, or some other physiological explanation? Either way, consider explaining this result with a bit more detail, given the striking pattern.

275...I'm not sure that I see the clear connection between the results and this conclusion. Figure 3b shows seasonal patterns of similar magnitude in bone and skin for $\delta^{13}\text{C}$, and if anything, a stronger seasonal pattern in bone than in skin for $\delta^{15}\text{N}$. Couldn't one make the argument that these results indicate that, at least in young animals, bone can be just as seasonal as skin? That is illogical however based on prior knowledge of turnover rates, so another explanation is needed.

How much of this effect reflects the use of data from reference 19 (eg, is there something consistently different about the diet and/or season of animals in reference 19 that could explain this trend)?

In addition, a difference in turnover rate between bone and skin would not seem to explain a population-wide trend towards skin having a lower $\delta^{13}\text{C}$ and a higher $\delta^{15}\text{N}$ than bone, unless animals were all consistently sampled during one season in which the diet had recently switched to something with lower $\delta^{13}\text{C}$ and higher $\delta^{15}\text{N}$. If you are sampling across seasons, shouldn't the difference in turnover rate instead cause the skin-bone offset to have an average near zero but with greater variation? In other words, if the explanation was just turnover rate, shouldn't the black trend lines and the dotted trend lines in Figure 3a basically overlap, with just a wide spread of individual data points?

When I look at Figure 3, I am actually more tempted to draw the conclusion that there is some kind of consistent routing of carbon and nitrogen that differs between skin and bone, which results in (all other things being equal) skin that will tend to have lower $\delta^{13}\text{C}$ and higher $\delta^{15}\text{N}$ than bone. Of course, that would be unlikely to reflect protein composition or amino acid composition, as the authors already addressed those factors. Is there any chance that the ultimate source of the amino acids can differ between these tissues? Perhaps some "on-site" synthesis of non-essential amino acids occurs in these tissues, using carbon and nitrogen that have been sourced via different biochemical pathways?

Review form: Reviewer 2

Is the manuscript scientifically sound in its present form?

Yes

Are the interpretations and conclusions justified by the results?

Yes

Is the language acceptable?

Yes

Do you have any ethical concerns with this paper?

No

Have you any concerns about statistical analyses in this paper?

No

Recommendation?

Major revision is needed (please make suggestions in comments)

Comments to the Author(s)

Review of "A modern baseline for the paired isotopic ($\delta^{13}\text{C}$ and $\delta^{15}\text{N}$) analysis of skin and bone in terrestrial mammals" by Doherty et al.

This manuscript measures the difference in $\delta^{13}\text{C}$ and $\delta^{15}\text{N}$ values between bone and skin collagen in domestic sheep and pigs. The rationale is that while the stable isotopic compositions of either skin or bone collagen can be used for dietary reconstructions, they each integrate different timescales so that if used in combination, a better dietary picture can be gleaned in archeological studies. This manuscript is well written, straight forward, and to the point. I very

much enjoyed reading it. The only criticism I have is that this study is rather specialized in that it necessarily deals with a very controlled experiment. In this regard, it may not be completely applicable when dealing with archeological studies whose main focus is usually ancient humans and whose life histories are unknown. I also note some analytical problems that need to be clarified or addressed in the manuscript. Nevertheless, if one assumes that the physiology of mammals are similar, this study may help set the baseline for studies that need to incorporate the stable isotopic compositions of both skin and bone collagen. In this respect, I think this is a worthwhile contribution and should be published with revisions as indicated below.

Stable isotope analyses:

What references were used for calculation of $\delta^{13}\text{C}$ and $\delta^{15}\text{N}$ values? The authors state that they calibrated their analytical runs with USGS40. This single-point calibration is not only unusual, but may result in significant error. This is especially the case for values that are far away from the single-point reference value, which certainly is the case here for nitrogen. USGS40 has $\delta^{13}\text{C}$ and $\delta^{15}\text{N}$ values of -26 and -4.5 per mil respectively. Stable isotope laboratories should use a two-point calibration whereby they calibrate a set of reference materials to VPDB and AIR (usually with USGS40 and USGS41), and then use those in-house references in their analytical runs. Best practices (ie Bond and Hobson, 2012) indicate the authors should use a set of reference materials that span the range of the unknowns. The authors state they use three in-house references for uncertainty measurement and I wonder if instead they meant they measure their unknowns against these references rather than USGS40. At any rate, it is not clear and the the calibrated delta values of their references should be reported.

L.204. The p-value to measure the significance of the differences between sheep and pig skin-bone for $\delta^{13}\text{C}$ values is 0.042. This is very close to the usual threshold of 0.05. Therefore, the sentence "significantly more depleted" may be overstating reality here.

Figures. I suggest the authors remove Fig 1 and substitute Fig 3 - see below.

Figure 1. I dont see where this came from at all. The amino acid makeup of skin and bone from bovine (cow) tissue? Is this figure taken from the study referenced in the text? At any rate it seems out of place and serves only to illustrate the citation. I suggest the authors omit this figure and just leave the referenced statement.

Figure 2. The authors present a comparison of skin-bone differences from the literature including the present study. Moreover, I think this brings up an important aspect - how do these results compare to those of previous investigations? From the figure, it seems that previous investigations found skin-bone differences that range from -2 to +1 per mil for carbon and 0 to +5 per mil or so for nitrogen isotopes. Why would this study be a more accurate or reliable measurement of skin-bone differences than those in the literature? The authors touch on this in the introduction, but this should be expanded and moved to the discussion. It should follow after the results of the study. A paragraph or three discussing this would add considerably to the manuscript.

Figure 3a. The error envelope of the regression lines should be included on this plot.

Review form: Reviewer 3

Is the manuscript scientifically sound in its present form?

Yes

Are the interpretations and conclusions justified by the results?

No

Is the language acceptable?

Yes

Do you have any ethical concerns with this paper?

No

Have you any concerns about statistical analyses in this paper?

Yes

Recommendation?

Major revision is needed (please make suggestions in comments)

Comments to the Author(s)

The authors present a relatively small dataset examining carbon and nitrogen isotope discrimination between bone and skin collagen gathered from pigs and sheep. The study provides some useful information to help guide the interpretation of archaeological collagen samples (e.g., isotopic offsets between skin and bone). However, after reading paper I am left very confused over the statistical design and reporting of statistical tests. For example, there are many cases where statistical tests are only partly reported, and in some cases, not reported at all. I provide general and more specific line-by-line comments below.

General Comments

1. My major concern relates to the authors statistical design and reporting of statistics. Even for basic pairwise comparisons the test statistics should be reported in addition to p values. I also found reported results that were not described in the statistical methods. For example, the correlation analyses used to examine big delta15N skin-bone and age. I also see that linear regressions are reported on figures, but never mentioned in the methods?
2. I also think the authors could potentially use a more integrative statistical design, for example generalized linear models, or linear mixed effect models, where the effects of predictor variables can be tested within a holistic modelling framework.
3. At the end of their introduction, the authors should improve the rationale behind their study, more-so than just 'expanding baseline data'. There needs to be a greater novelty component to excite the reader. Additionally, the authors consistently attack specific published work for not following appropriate QAQC procedures. I would caution against the way these caveats are written and feel that the authors can make their arguments effectively without writing a paragraph about specific papers that (in their opinion) are not up to scratch (there are a lot!).
4. The 'summary' does a poor job of explaining the study rationale or discussing the general implications the findings have for investigating biological/physiological processes in extant of archaeological settings.

Specific comments

Line 19 – Depletion of 13C usually results in a lower (more negative) isotope value, so saying depleted by -0.7 permille reads like a contraction.

Line 31 – Life-histories is a very general term that describes many biological and physiological processes in animals. I would be more specific here.

Line 33 – Metabolically inert tissues can reveal both long and short-term changes to diet, habitat use, and physiological condition, so I don't think these tissues are an appropriate example here.

Line 38 – I would define isotopic discrimination here.

Lines 55 – 58 – This is a good rationale that needs to be integrated into the summary section of the manuscript.

Line 59 – I would argue that all contaminants referred to here can be considered ‘exogenous’. Consider deleting.

Line 63 – The authors are missing a recent publication by Guiry and Szpak (2020), that I suggest citing here.

Guiry, E. J., & Szpak, P. (2020). Quality control for modern bone collagen stable carbon and nitrogen isotope measurements. *Methods in Ecology and Evolution*, 11(9), 1049-1060.

Lines 74 – 91 – I appreciate the authors message here but would caution against the undressing of other published papers so directly. They can effectively make the point about a lack of pre-treatment and cite a few papers where this is the case.

Lines 95 – 97 – Again, you can make these points without so directly calling out the work of others. This sentence repeats much of lines 74 – 91.

Line 167 – I think you mean that data did not conform to a normal distribution, thus the resultant statistical tests were non-parametric in nature.

Line 187 – Please report the full results of pairwise comparisons and not just p values.

Lines 192 – 202 – This is all interpretation and belongs in the discussion, not the results.

Lines 205 – 209 – Again this is discussion and not results.

Lines 219 – 220 – You cannot just remove data without a rationale and statistically grounded justification.

Line 228 – I don’t understand what you mean by ‘turnover discrimination’. Consider rephrasing.

Lines 234 – 236 – You need to report $\delta^{13}\text{C}$ and $\delta^{15}\text{N}$ values to guide the reader through the directionality of the seasonal effects. Also which statistical analyses indicated differences across season?

Line 253 – 254 – Results of statistical analyses are not reported.

Decision letter (RSOS-210316.R0)

Dear Dr Doherty

The Editors assigned to your paper RSOS-210316 "A modern baseline for the paired isotopic ($\delta^{13}\text{C}$ and $\delta^{15}\text{N}$) analysis of skin and bone in terrestrial mammals" have made a decision based on their reading of the paper and any comments received from reviewers.

Regrettably, in view of the reports received, the manuscript has been rejected in its current form. However, a new manuscript may be submitted which takes into consideration these comments.

Whilst each reviewer saw positive aspects of the paper and recommended 'major revision' rather than 'reject', taken together the major revisions requested are rather substantial. Both I and the Associate Editor are of the opinion that the way in which Royal Society Open Science operates, not allowing repeated revisions, would be too constraining for you in offering scope for revision to meet the various demands of the referees. We hope that the decision of 'rejection, allowing resubmission' will give you a more realistic opportunity to bring your interesting paper to a form that will be appropriate for publication in this journal.

Additionally, please accept our sincere apologies for the unusual delays incurred during the review process. We regret that it proved more difficult than usual to acquire referees for your paper, and Editor and staff absences related to the pandemic also caused some delays. We will endeavour to do all that we can to expedite your paper once you have submitted a revised version.

We invite you to respond to the comments supplied below and prepare a resubmission of your manuscript. Below the referees' and Editors' comments (where applicable) we provide additional requirements. We provide guidance below to help you prepare your revision.

Please note that resubmitting your manuscript does not guarantee eventual acceptance, and we do not generally allow multiple rounds of revision and resubmission, so we urge you to make every effort to fully address all of the comments at this stage. If deemed necessary by the Editors, your manuscript will be sent back to one or more of the original reviewers for assessment. If the original reviewers are not available, we may invite new reviewers.

Please resubmit your revised manuscript and required files (see below) no later than 05-Jan-2022. Note: the ScholarOne system will 'lock' if resubmission is attempted on or after this deadline. If you do not think you will be able to meet this deadline, please contact the editorial office immediately.

Please note article processing charges apply to papers accepted for publication in Royal Society Open Science (<https://royalsocietypublishing.org/rsos/charges>). Charges will also apply to papers transferred to the journal from other Royal Society Publishing journals, as well as papers submitted as part of our collaboration with the Royal Society of Chemistry (<https://royalsocietypublishing.org/rsos/chemistry>). Fee waivers are available but must be requested when you submit your manuscript (<https://royalsocietypublishing.org/rsos/waivers>).

Thank you for submitting your manuscript to Royal Society Open Science and we look forward to receiving your resubmission. If you have any questions at all, please do not hesitate to get in touch.

on behalf of Professor Marcelo Sanchez (Associate Editor) and Peter Haynes (Subject Editor)
openscience@royalsociety.org

Associate Editor Comments to Author (Professor Marcelo Sanchez):

There is much potential in this study if the reviews are carefully considered

Reviewer comments to Author:

Reviewer: 1

Comments to the Author(s)

Although I have selected "Major revision", I want to be clear that overall, I think this is a high-quality paper that will make a worthwhile contribution.

I base my response only on the fact that, unless I am misinterpreting the data, I think that the authors could dig deeper into their data and provide a more nuanced explanation for the trends

that they observe, and perhaps even state that their findings appear to challenge some of the fundamental principles that guided their experimental design. In the event that I am wrong, my input should at least provide the authors guidance for where to clarify their text. Details are below.

Line 82...this may be a good place to describe that lipid contamination, generally indicated by high CN ratios, often results in reduced d13C values because of the carbon fractionation that occurs during lipid synthesis as described by DeNiro and Epstein.

88...prior to these passages, the authors have only described the processes of lipid extraction and filtration (lines 65-66). Here, the authors need to define or describe gelatinization, freeze-drying, and demineralization.

98...see previous comment about line 88.

175...is this yield based on dry mass?

178...I believe this should be in past tense?

198...change to "...the elevated skin d15N values observed..."

202...rephrase this to be more precise. It is clear from modern animals that an enrichment of at least 1.1 per mil in the skin, relative to bone, can occur even in the absence of any diagenetic explanation.

231...Lübcker et al (2020. Conservation Physiology DOI: 10.1093/conphys/coaa060) conducted a relevant analysis, and offer insight into the isotope dynamics of specific amino acids during allocation from mothers to fetal tissue. Consider including their work into this discussion.

246...the seasonal variation in the d13C of bone is small, with medians ranging only from -24 to -25. However, the seasonal variation in the d15N of bone is surprisingly large, with medians from 6 to 8. Given that d15N increases by roughly 3 per mil with a trophic level (an approximation, but a useful one), this change in bone represents nearly a trophic level. This is unexpected, because as described in the manuscript, bone should be a long-term average. However, nearly all of the sheep were less than one year of age; perhaps they had not yet lived through enough seasons for the bone to become more homogenous over time. Or perhaps in the first year of life, bone is still more of a reservoir for stored energy, or some other physiological explanation? Either way, consider explaining this result with a bit more detail, given the striking pattern.

275...I'm not sure that I see the clear connection between the results and this conclusion. Figure 3b shows seasonal patterns of similar magnitude in bone and skin for d13C, and if anything, a stronger seasonal pattern in bone than in skin for d15N. Couldn't one make the argument that these results indicate that, at least in young animals, bone can be just as seasonal as skin? That is illogical however based on prior knowledge of turnover rates, so another explanation is needed. How much of this effect reflects the use of data from reference 19 (eg, is there something consistently different about the diet and/or season of animals in reference 19 that could explain this trend)?

In addition, a difference in turnover rate between bone and skin would not seem to explain a population-wide trend towards skin having a lower d13C and a higher d15N than bone, unless animals were all consistently sampled during one season in which the diet had recently switched to something with lower d13C and higher d15N. If you are sampling across seasons, shouldn't the difference in turnover rate instead cause the skin-bone offset to have an average near zero but

with greater variation? In other words, if the explanation was just turnover rate, shouldn't the black trend lines and the dotted trend lines in Figure 3a basically overlap, with just a wide spread of individual data points?

When I look at Figure 3, I am actually more tempted to draw the conclusion that there is some kind of consistent routing of carbon and nitrogen that differs between skin and bone, which results in (all other things being equal) skin that will tend to have lower $\delta^{13}\text{C}$ and higher $\delta^{15}\text{N}$ than bone. Of course, that would be unlikely to reflect protein composition or amino acid composition, as the authors already addressed those factors. Is there any chance that the ultimate source of the amino acids can differ between these tissues? Perhaps some "on-site" synthesis of non-essential amino acids occurs in these tissues, using carbon and nitrogen that have been sourced via different biochemical pathways?

Reviewer: 2

Comments to the Author(s)

Review of "A modern baseline for the paired isotopic ($\delta^{13}\text{C}$ and $\delta^{15}\text{N}$) analysis of skin and bone in terrestrial mammals" by Doherty et al.

This manuscript measures the difference in $\delta^{13}\text{C}$ and $\delta^{15}\text{N}$ values between bone and skin collagen in domestic sheep and pigs. The rationale is that while the stable isotopic compositions of either skin or bone collagen can be used for dietary reconstructions, they each integrate different timescales so that if used in combination, a better dietary picture can be gleaned in archeological studies. This manuscript is well written, straight forward, and to the point. I very much enjoyed reading it. The only criticism I have is that this study is rather specialized in that it necessarily deals with a very controlled experiment. In this regard, it may not be completely applicable when dealing with archeological studies whose main focus is usually ancient humans and whose life histories are unknown. I also note some analytical problems that need to be clarified or addressed in the manuscript. Nevertheless, if one assumes that the physiology of mammals are similar, this study may help set the baseline for studies that need to incorporate the stable isotopic compositions of both skin and bone collagen. In this respect, I think this is a worthwhile contribution and should be published with revisions as indicated below.

Stable isotope analyses:

What references were used for calculation of $\delta^{13}\text{C}$ and $\delta^{15}\text{N}$ values? The authors state that they calibrated their analytical runs with USGS40. This single-point calibration is not only unusual, but may result in significant error. This is especially the case for values that are far away from the single-point reference value, which certainly is the case here for nitrogen. USGS40 has $\delta^{13}\text{C}$ and $\delta^{15}\text{N}$ values of -26 and -4.5 per mil respectively. Stable isotope laboratories should use a two-point calibration whereby they calibrate a set of reference materials to VPDB and AIR (usually with USGS40 and USGS41), and then use those in-house references in their analytical runs. Best practices (ie Bond and Hobson, 2012) indicate the authors should use a set of reference materials that span the range of the unknowns. The authors state they use three in-house references for uncertainty measurement and I wonder if instead they meant they measure their unknowns against these references rather than USGS40. At any rate, it is not clear and the the calibrated delta values of their references should be reported.

L.204. The p-value to measure the significance of the differences between sheep and pig skin-bone for $\delta^{13}\text{C}$ values is 0.042. This is very close to the usual threshold of 0.05. Therefore, the sentence "significantly more depleted" may be overstating reality here.

Figures. I suggest the authors remove Fig 1 and substitute Fig 3 - see below.

Figure 1. I don't see where this came from at all. The amino acid makeup of skin and bone from bovine (cow) tissue? Is this figure taken from the study referenced in the text? At any rate it seems out of place and serves only to illustrate the citation. I suggest the authors omit this figure and just leave the referenced statement.

Figure 2. The authors present a comparison of skin-bone differences from the literature including the present study. Moreover, I think this brings up an important aspect - how do these results compare to those of previous investigations? From the figure, it seems that previous investigations found skin-bone differences that range from -2 to +1 per mil for carbon and 0 to +5 per mil or so for nitrogen isotopes. Why would this study be a more accurate or reliable measurement of skin-bone differences than those in the literature? The authors touch on this in the introduction, but this should be expanded and moved to the discussion. It should follow after the results of the study. A paragraph or three discussing this would add considerably to the manuscript.

Figure 3a. The error envelope of the regression lines should be included on this plot.

Reviewer: 3

Comments to the Author(s)

The authors present a relatively small dataset examining carbon and nitrogen isotope discrimination between bone and skin collagen gathered from pigs and sheep. The study provides some useful information to help guide the interpretation of archaeological collagen samples (e.g., isotopic offsets between skin and bone). However, after reading paper I am left very confused over the statistical design and reporting of statistical tests. For example, there are many cases where statistical tests are only partly reported, and in some cases, not reported at all. I provide general and more specific line-by-line comments below.

General Comments

1. My major concern relates to the authors' statistical design and reporting of statistics. Even for basic pairwise comparisons the test statistics should be reported in addition to p values. I also found reported results that were not described in the statistical methods. For example, the correlation analyses used to examine $\delta^{15}\text{N}$ skin-bone and age. I also see that linear regressions are reported on figures, but never mentioned in the methods?

2. I also think the authors could potentially use a more integrative statistical design, for example generalized linear models, or linear mixed effect models, where the effects of predictor variables can be tested within a holistic modelling framework.

3. At the end of their introduction, the authors should improve the rationale behind their study, more-so than just 'expanding baseline data'. There needs to be a greater novelty component to excite the reader. Additionally, the authors consistently attack specific published work for not following appropriate QAQC procedures. I would caution against the way these caveats are written and feel that the authors can make their arguments effectively without writing a paragraph about specific papers that (in their opinion) are not up to scratch (there are a lot!).

4. The 'summary' does a poor job of explaining the study rationale or discussing the general implications the findings have for investigating biological/physiological processes in extant of archaeological settings.

Specific comments

Line 19 - Depletion of ^{13}C usually results in a lower (more negative) isotope value, so saying depleted by -0.7 permille reads like a contraction.

Line 31 - Life-histories is a very general term that describes many biological and physiological processes in animals. I would be more specific here.

Line 33 – Metabolically inert tissues can reveal both long and short-term changes to diet, habitat use, and physiological condition, so I don't think these tissues are an appropriate example here.

Line 38 – I would define isotopic discrimination here.

Lines 55 – 58 – This is a good rationale that needs to be integrated into the summary section of the manuscript.

Line 59 – I would argue that all contaminants referred to here can be considered 'exogenous'. Consider deleting.

Line 63 – The authors are missing a recent publication by Guiry and Szpak (2020), that I suggest citing here.

Guiry, E. J., & Szpak, P. (2020). Quality control for modern bone collagen stable carbon and nitrogen isotope measurements. *Methods in Ecology and Evolution*, 11(9), 1049-1060.

Lines 74 – 91 – I appreciate the authors message here but would caution against the undressing of other published papers so directly. They can effectively make the point about a lack of pre-treatment and cite a few papers where this is the case.

Lines 95 – 97 – Again, you can make these points without so directly calling out the work of others. This sentence repeats much of lines 74 – 91.

Line 167 – I think you mean that data did not conform to a normal distribution, thus the resultant statistical tests were non-parametric in nature.

Line 187 – Please report the full results of pairwise comparisons and not just p values.

Lines 192 – 202 – This is all interpretation and belongs in the discussion, not the results.

Lines 205 – 209 – Again this is discussion and not results.

Lines 219 – 220 – You cannot just remove data without a rationale and statistically grounded justification.

Line 228 – I don't understand what you mean by 'turnover discrimination'. Consider rephrasing.

Lines 234 – 236 – You need to report d13C and d15N values to guide the reader through the directionality of the seasonal effects. Also which statistical analyses indicated differences across season?

Line 253 – 254 – Results of statistical analyses are not reported.

===PREPARING YOUR MANUSCRIPT===

===PREPARING YOUR REVISION IN SCHOLARONE===

Author's Response to Decision Letter for (RSOS-210316.R0)

See Appendix A.

RSOS-211587.R0

Review form: Reviewer 1

Is the manuscript scientifically sound in its present form?

Yes

Are the interpretations and conclusions justified by the results?

Yes

Is the language acceptable?

Yes

Do you have any ethical concerns with this paper?

No

Have you any concerns about statistical analyses in this paper?

Yes

Recommendation?

Accept with minor revision (please list in comments)

Comments to the Author(s)

The authors have done a thorough and admirable job addressing my comments.

My only input at this stage is to correct the grammar for the statement below, and to consider two relevant citations for it:

"We instead postulate the consistent routing of carbon and nitrogen that differs between skin and bone, with potentially on-site synthesis of non-essential amino acids occurring carbon and nitrogen that have been sourced via different biochemical pathways."

Citations:

Jim et al. 2006. Quantifying dietary macronutrient sources of carbon for bone collagen biosynthesis using natural abundance stable carbon isotope analysis. *British J Nutrition* 95:1055-1062

Newsome et al. 2014. Amino acid $\delta^{13}\text{C}$ analysis shows flexibility in the routing of dietary protein and lipids to the tissue of an omnivore. *Integr and Comp Biol* 54:890-902.

Decision letter (RSOS-211587.R0)

Dear Dr Doherty

On behalf of the Editors, we are pleased to inform you that your Manuscript RSOS-211587 "A modern baseline for the paired isotopic analysis of skin and bone in terrestrial mammals" has been accepted for publication in Royal Society Open Science subject to minor revision in accordance with the referees' reports. Please find the referees' comments along with any feedback from the Editors below my signature.

Please submit your revised manuscript and required files (see below) no later than 7 days from today's (ie 10-Dec-2021) date. Note: the ScholarOne system will 'lock' if submission of the revision is attempted 7 or more days after the deadline. If you do not think you will be able to meet this deadline please contact the editorial office immediately.

on behalf of Professor Marcelo Sanchez (Associate Editor) and Peter Haynes (Subject Editor)
openscience@royalsociety.org

Reviewer comments to Author:

Reviewer: 1

Comments to the Author(s)

The authors have done a thorough and admirable job addressing my comments.

My only input at this stage is to correct the grammar for the statement below, and to consider two relevant citations for it:

"We instead postulate the consistent routing of carbon and nitrogen that differs between skin and bone, with potentially on-site synthesis of non-essential amino acids occurring carbon and nitrogen that have been sourced via different biochemical pathways."

Citations:

Jim et al. 2006. Quantifying dietary macronutrient sources of carbon for bone collagen biosynthesis using natural abundance stable carbon isotope analysis. *British J Nutrition* 95:1055-1062

Newsome et al. 2014. Amino acid $\delta^{13}\text{C}$ analysis shows flexibility in the routing of dietary protein and lipids to the tissue of an omnivore. *Integr and Comp Biol* 54:890-902.

===PREPARING YOUR MANUSCRIPT===

one version should clearly identify all the changes that have been made (for instance, in coloured highlight, in bold text, or tracked changes);

===PREPARING YOUR REVISION IN SCHOLARONE===

-- If you are requesting an article processing charge waiver, you must select the relevant waiver option (if requesting a discretionary waiver, the form should have been uploaded, see 'File upload' above).

-- If you have uploaded any electronic supplementary (ESM) files, please ensure you follow the guidance at <https://royalsociety.org/journals/authors/author-guidelines/#supplementary-material> to include a suitable title and informative caption. An example of appropriate titling and captioning may be found at https://figshare.com/articles/Table_S2_from_Is_there_a_trade-off_between_peak_performance_and_performance_breadth_across_temperatures_for_aerobic_scope_in_teleost_fishes_/3843624.

Author's Response to Decision Letter for (RSOS-211587.R0)

See Appendix B.

Decision letter (RSOS-211587.R1)

Dear Dr Doherty,

I am pleased to inform you that your manuscript entitled "A modern baseline for the paired isotopic analysis of skin and bone in terrestrial mammals" is now accepted for publication in Royal Society Open Science.

Please see the Royal Society Publishing guidance on how you may share your accepted author manuscript at <https://royalsociety.org/journals/ethics-policies/media-embargo/>. After publication, some additional ways to effectively promote your article can also be found here

<https://royalsociety.org/blog/2020/07/promoting-your-latest-paper-and-tracking-your-results/>.

on behalf of Professor Marcelo Sanchez (Associate Editor) and Peter Haynes (Subject Editor)
openscience@royalsociety.org

Appendix A

Author's Response to Reviewers Comments

Reviewer: 1

Although I have selected "Major revision", I want to be clear that overall, I think this is a high-quality paper that will make a worthwhile contribution.

I base my response only on the fact that, unless I am misinterpreting the data, I think that the authors could dig deeper into their data and provide a more nuanced explanation for the trends that they observe, and perhaps even state that their findings appear to challenge some of the fundamental principles that guided their experimental design. In the event that I am wrong, my input should at least provide the authors guidance for where to clarify their text. Details are below.

Line 82...this may be a good place to describe that lipid contamination, generally indicated by high CN ratios, often results in reduced $\delta^{13}\text{C}$ values because of the carbon fractionation that occurs during lipid synthesis as described by DeNiro and Epstein.

- We have added the following sentence:

“elevated C:N ratios (>3.5) indicative of the presence of carbon-rich contaminants, such as lipids, which may artificially reduce $\delta^{13}\text{C}$ values due to the carbon fractionation that occurs during lipid synthesis [36].” (Lines 58-60)

88...prior to these passages, the authors have only described the processes of lipid extraction and filtration (lines 65-66). Here, the authors need to define or describe gelatinization, freeze-drying, and demineralization.

- We have amended the sentence, and now better defines these steps:

“...undergone pretreatment (solvent extraction of lipids; demineralisation in HCl acid; gelatinisation of the acid-insoluble fraction in pH 3 water; filtration and freeze-drying) and produced...” (Lines 61-63)

98...see previous comment about line 88.

- Following the recommendation of Reviewer 3, the section which extensively critiqued other studies methodology has now been reduced to:

“Many archaeological skin samples fail to meet these quality indicators due to analysis inadequate pretreatment, often producing elevated C:N ratios (>3.5)” (Line 56-58)

175...is this yield based on dry mass?

- Yes, dry mass, which is now specified in the text (Line 131)

178...I believe this should be in past tense?

- “ranges” has now been amended to “ranged” (Line 133), and similar instances in the results section have been changed to past tense.

198...change to “...the elevated skin $\delta^{15}\text{N}$ values observed...”

- Amended (Line 162)

202...rephrase this to be more precise. It is clear from modern animals that an enrichment of at least 1.1 per mil in the skin, can occur even in the absence of any diagenetic explanation.

- For clarity we have added the following sentence. The largest enrichment seen in our modern animals is 2.9‰:

“While this has the potential to influence the nitrogen isotope values, it is clear from modern animals that an enrichment of at least 2.9‰ in the skin, relative to bone, can occur even in the absence of any diagenetic explanation.” (Line 164-6)

231...Lübcker et al (2020. Conservation Physiology DOI: 10.1093/conphys/coaa060) conducted a relevant analysis, and offer insight into the isotope dynamics of specific amino acids during allocation from mothers to fetal tissue. Consider including their work into this discussion.

- We thank the reviewer for making us aware of this publication.

246...the seasonal variation in the d13C of bone is small, with medians ranging only from -24 to -25. However, the seasonal variation in the d15N of bone is surprisingly large, with medians from 6 to 8. Given that d15N increases by roughly 3 per mil with a trophic level (an approximation, but a useful one), this change in bone represents nearly a trophic level. This is unexpected, because as described in the manuscript, bone should be a long-term average. However, nearly all of the sheep were less than one year of age; perhaps they had not yet lived through enough seasons for the bone to become more homogenous over time. Or perhaps in the first year of life, bone is still more of a reservoir for stored energy, or some other physiological explanation? Either way, consider explaining this result with a bit more detail, given the striking pattern.

275...I'm not sure that I see the clear connection between the results and this conclusion. Figure 3b shows seasonal patterns of similar magnitude in bone and skin for d13C, and if anything, a stronger seasonal pattern in bone than in skin for d15N. Couldn't one make the argument that these results indicate that, at least in young animals, bone can be just as seasonal as skin? That is illogical however based on prior knowledge of turnover rates, so another explanation is needed. How much of this effect reflects the use of data from reference 19 (eg, is there something consistently different about the diet and/or season of animals in reference 19 that could explain this trend)?

- On further analysis, the Reviewer is correct that much of the trends seen in bone isotope values are influenced by reference 19 (von Holstein *et al.* 2013). Consequently, we have removed much of the discussion on seasonal variation in isotope values.

In addition, a difference in turnover rate between bone and skin would not seem to explain a population-wide trend towards skin having a lower d13C and a higher d15N than bone, unless animals were all consistently sampled during one season in which the diet had recently switched to something with lower d13C and higher d15N. If you are sampling across seasons, shouldn't the difference in turnover rate instead cause the skin-bone offset to have an average near zero but with greater variation? In other words, if the explanation was just turnover rate, shouldn't the black trend lines and the dotted trend lines in Figure 3a basically overlap, with just a wide spread of individual data points?

When I look at Figure 3, I am actually more tempted to draw the conclusion that there is some kind of consistent routing of carbon and nitrogen that differs between skin and bone, which results in (all other things being equal) skin that will tend to have lower d13C and higher d15N than bone. Of course, that would be unlikely to reflect protein composition or amino acid composition, as the authors already addressed those factors. Is there any chance that the ultimate source of the amino acids can differ between these tissues? Perhaps some "on-site" synthesis of non-essential amino acids occurs in these tissues, using carbon and nitrogen that have been sourced via different biochemical pathways?

- We thank the Reviewer for this insightful and detailed comment. We agree that our original submission placed too greater emphasis on the role of different turnover rates on inter-tissue isotopic variation. As the Reviewer indicates, the consistent offset in both $\delta^{13}\text{C}$ and $\delta^{15}\text{N}$ is unlikely to be result of turnover alone. In light of this, we have reviewed and discussed our results and now present the following conclusion, drawing upon the Reviewer's expertise:

“Contrary to the assumptions made by previous studies, we reject the hypothesis that $\delta^{13}\text{C}$ and $\delta^{15}\text{N}$ variation between skin and bone collagen in an individual is the result of their

contrasting turnover rates. In this analysis, a variety of animals raised under different management practices, slaughtered in different years and at different ages, showed a population-wide trend for skin to have a lower $\delta^{13}\text{C}$ and higher $\delta^{15}\text{N}$ than bone. If the explanation was simply turnover rate, we would expect an average near zero, but with considerable variation. It is also unlikely to reflect protein or amino acid composition due to the minor variation between these tissues.

We instead postulate the consistent routing of carbon and nitrogen that differs between skin and bone, with potentially on-site synthesis of non-essential amino acids occurring carbon and nitrogen that have been sourced via different biochemical pathways. In the absence of secondary influences (i.e. dietary change, physiological stress or diagenesis), this results in skin having a lower $\delta^{13}\text{C}$ and higher $\delta^{15}\text{N}$ than bone.” (Lines 229-240)

Reviewer: 2

Comments to the Author(s)

Review of “A modern baseline for the paired isotopic ($\delta^{13}\text{C}$ and $\delta^{15}\text{N}$) analysis of skin and bone in terrestrial mammals” by Doherty et al.

This manuscript measures the difference in $\delta^{13}\text{C}$ and $\delta^{15}\text{N}$ values between bone and skin collagen in domestic sheep and pigs. The rationale is that while the stable isotopic compositions of either skin or bone collagen can be used for dietary reconstructions, they each integrate different timescales so that if used in combination, a better dietary picture can be gleaned in archeological studies. This manuscript is well written, straight forward, and to the point. I very much enjoyed reading it. The only criticism I have is that this study is rather specialized in that it necessarily deals with a very controlled experiment. In this regard, it may not be completely applicable when dealing with archeological studies whose main focus is usually ancient humans and whose life histories are unknown. I also note some analytical problems that need to be clarified or addressed in the manuscript. Nevertheless, if one assumes that the physiology of mammals are similar, this study may help set the baseline for studies that need to incorporate the stable isotopic compositions of both skin and bone collagen. In this respect, I think this is a worthwhile contribution and should be published with revisions as indicated below.

Stable isotope analyses:

What references were used for calculation of $\delta^{13}\text{C}$ and $\delta^{15}\text{N}$ values? The authors state that they calibrated their analytical runs with USGS40. This single-point calibration is not only unusual, but may result in significant error. This is especially the case for values that are far away from the single-point reference value, which certainly is the case here for nitrogen. USGS40 has $\delta^{13}\text{C}$ and $\delta^{15}\text{N}$ values of -26 and -4.5 per mil respectively. Stable isotope laboratories should use a two-point calibration whereby they calibrate a set of reference materials to VPDB and AIR (usually with USGS40 and USGS41), and then use those in-house references in their analytical runs. Best practices (ie Bond and Hobson, 2012) indicate the authors should use a set of reference materials that span the range of the unknowns. The authors state they use three in-house references for uncertainty measurement and I wonder if instead they meant they measure their unknowns against these references rather than USGS40. At any rate, it is not clear and the the calibrated delta values of their references should be reported.

- We have clarified that, as Reviewer 2 presumed, a two-point calibration was used. The Methods section now states:

“Sample data were reported in standard delta per mil notation (δ ‰) relative to V-PDB ($\delta^{13}\text{C}$) and AIR ($\delta^{15}\text{N}$) international standards. Three laboratory reference materials were interspersed within the measurement run to correct for linearity and instrument drift. Each of the laboratory reference materials is checked regularly against international standards USGS40 and USGS41.” (Lines 115-118)

L.204. The p-value to measure the significance of the differences between sheep and pig skin-bone for $\delta^{13}\text{C}$ values is 0.042. This is very close to the usual threshold of 0.05. Therefore, the sentence “significantly more depleted” may be overstating reality here.

- The statement has been revised to:

“Mean $\Delta^{13}\text{C}_{(\text{skin-bone})}$ values in sheep are marginally more depleted than those in pigs ($t(28) = 2.1, P = 0.042$)”

Figures. I suggest the authors remove Fig 1 and substitute Fig 3 - see below.

Figure 1. I don't see where this came from at all. The amino acid makeup of skin and bone from bovine (cow) tissue? Is this figure taken from the study referenced in the text? At any rate it seems out of place and serves only to illustrate the citation. I suggest the authors omit this figure and just leave the referenced statement.

- Figure 1 was not taken from the study referenced in the text, but was a visualisation of data presented therein. However, we agree that the citation alone is sufficient and the figure has been removed.

Figure 2. The authors present a comparison of skin-bone differences from the literature including the present study. Moreover, I think this brings up an important aspect - how do these results compare to those of previous investigations? From the figure, it seems that previous investigations found skin-bone differences that range from -2 to +1 per mil for carbon and 0 to +5 per mil or so for nitrogen isotopes. Why would this study be a more accurate or reliable measurement of skin-bone differences than those in the literature? The authors touch on this in the introduction, but this should be expanded and moved to the discussion. It should follow after the results of the study. A paragraph or three discussing this would add considerably to the manuscript.

- The benefit of determining the $\Delta_{(\text{skin-bone})}$ offset in modern animals is due to the range of unknown cultural, environmental or physiological factors which may have influenced the value in archaeological populations. At present, the lack of baseline data from individuals with known life histories means that archaeological data is interpreted in comparison with other data. To emphasise this, the manuscript now states:

“At present, the interpretation of inter-tissue variation ($\Delta(\text{skin-bone})$) in mummified remains is based on comparisons with other archaeological material, which have attributed divergence to their contrasting turnover rates, with rapidly remodelling skin collagen incorporating alterations in environmental, cultural and physiological conditions in the months prior to death. While plausible, the lack of baseline data from individuals with known life histories has hindered evaluation of the explanations presented.” (Abstract, Lines 19-24)

and,

“While these interpretations are plausible, the lack of baseline data from individuals with known life histories confounds evaluation of the explanations presented. $\Delta(\text{skin-bone})$ discrimination in archaeological populations may be additionally influenced by differential decomposition patterns [26] and the presence of exogenous contaminants [27–29].” (Introduction, Lines 47-51)

and,

“This study provides an important dataset for the interpretation of skin-bone isotopic discrimination in archaeological studies, reporting the range of values that can be expected in the absence of geographical or dietary change, and diagenesis.” (Conclusion, Lines 242-244)

Figure 3a. The error envelope of the regression lines should be included on this plot.

- This Figure (now Figure 1) now includes a 95% CI.

Reviewer: 3

Comments to the Author(s)

The authors present a relatively small dataset examining carbon and nitrogen isotope discrimination between bone and skin collagen gathered from pigs and sheep. The study provides some useful information to help guide the interpretation of archaeological collagen samples (e.g., isotopic offsets between skin and bone). However, after reading paper I am left very confused over the statistical design and reporting of statistical tests. For example, there are many cases where statistical tests are only partly reported, and in some cases, not reported at all. I provide general and more specific line-by-line comments below.

General Comments

1. My major concern relates to the authors statistical design and reporting of statistics. Even for basic pairwise comparisons the test statistics should be reported in addition to p values. I also found reported results that were not described in the statistical methods. For example, the correlation analyses used to examine big delta¹⁵N skin-bone and age. I also see that linear regressions are reported on figures, but never mentioned in the methods?

- We thank the Reviewer 3 for their useful comment. In response, full test statistics are reported in the text. For example:

“...Mean $\Delta^{13}\text{C}_{(\text{skin-bone})}$ values in sheep are marginally more depleted than those in pigs ($t(28) = 2.1$, $P = 0.042$) although not for $\Delta^{15}\text{N}_{(\text{skin-bone})}$ values ($t(28) = -0.85$, $P = 0.41$).” (Lines 177-178).

All statistical tests used are now mentioned in the methods section, which reads:

“Statistical testing was carried out using the IBM SPSS Statistics 26 software package. Shapiro-Wilks test for normality indicated $\delta^{13}\text{C}$ and $\delta^{15}\text{N}$ data did not conform to a normal distribution ($P < 0.01$), thus the resultant statistical tests were non-parametric in nature. Significance of differences between bone and skin values determined using a Wilcoxon signed-rank test for paired samples. Significance of differences in $\Delta_{(\text{skin-bone})}$ values between species, sexes and health-status groups was determined using an Independent t-test. The correlation between age and $\Delta_{(\text{skin-bone})}$ was determined through the Pearson correlation coefficient.” (Line 121-128)

2. I also think the authors could potentially use a more integrative statistical design, for example generalized linear models, or linear mixed effect models, where the effects of predictor variables can be tested within a holistic modelling framework.

- We thank the Reviewer for the suggestion, however as our study seeks “to test the null hypothesis that there is no significant difference between $\delta^{13}\text{C}$ and $\delta^{15}\text{N}$ values in skin and bone collagen within an individual” (Lines 64-65), we feel that the Wilcoxon signed-rank test for paired samples is an appropriate method to test, and in this case reject, the null hypothesis.

3. At the end of their introduction, the authors should improve the rationale behind their study, more-so than just ‘expanding baseline data’. There needs to be a greater novelty component to excite the reader. Additionally, the authors consistently attack specific published work for not following appropriate QAQC procedures. I would caution against the way these caveats are written and feel that the authors can make their arguments effectively without writing a paragraph about specific papers that (in their opinion) are not up to scratch (there are a lot!).

- In light of the Reviewer's comments we have substantially reduced our critique of published studies. This section now reads:

“Many archaeological skin samples fail to meet these quality indicators due to inadequate pretreatment, often producing elevated C:N ratios (>3.5) indicative of the presence of carbon-rich contaminants, such as lipids, which may artificially reduce $\delta^{13}\text{C}$ values due to the carbon fractionation that occurs during lipid synthesis [36]. Consequently, the only modern mammalian skin sample known to have undergone pretreatment (solvent extraction of lipids; demineralisation in HCl acid; gelatinisation of the acid-insoluble fraction in pH 3 water; filtration and freeze-drying) and produced an acceptable C:N ratio (3.3) comes from a single pig [28].” (Lines 56-63)

4. The ‘summary’ does a poor job of explaining the study rationale or discussing the general implications the findings have for investigating biological/physiological processes in extant of archaeological settings.

- We have amended the summary to include the rationale that at present, interpretation of inter-tissue spacing in archaeological material is made in comparison with other archaeological samples, despite the multitude of unknown factors that may have influenced these values. We go on to highlight how the analysis of modern animals raised under a variety of management practices provides a way of exploring the spacing seen in a population without geographical, dietary or diagenetic change. (see comment below).

Specific comments

Line 19 – Depletion of ^{13}C usually results in a lower (more negative) isotope value, so saying depleted by -0.7 permille reads like a contraction.

- Amended

Line 31 – Life-histories is a very general term that describes many biological and physiological processes in animals. I would be more specific here.

Line 33 – Metabolically inert tissues can reveal both long and short-term changes to diet, habitat use, and physiological condition, so I don't think these tissues are an appropriate example here.

- Recommended sections deleted

Line 38 – I would define isotopic discrimination here.

- To define this, we now state:

“While variations in amino acid composition typically result in different tissues within the same individual displaying divergent isotopic values [20–22]...” (Line 38-39)

Lines 55 – 58 – This is a good rationale that needs to be integrated into the summary section of the manuscript.

- This has now been included in the summary section (see above)

Line 59 – I would argue that all contaminants referred to here can be considered ‘exogenous’. Consider deleting.

- Recommended sections deleted

Line 63 – The authors are missing a recent publication by Guiry and Szpak (2020), that I suggest citing here.

Guiry, E. J., & Szpak, P. (2020). Quality control for modern bone collagen stable carbon and nitrogen isotope measurements. *Methods in Ecology and Evolution*, 11(9), 1049-1060.

- We thank the Reviewer for making us aware of this publication. We have now cited it in this section [reference 33].

Lines 74 – 91 – I appreciate the authors message here but would caution against the undressing of other published papers so directly. They can effectively make the point about a lack of pre-treatment and cite a few papers where this is the case.

- As stated earlier, we have followed the Reviewer’s advice and substantially revised the review of other published papers.

Lines 95 – 97 – Again, you can make these points without so directly calling out the work of others. This sentence repeats much of lines 74 – 91.

- See above.

Line 167 – I think you mean that data did not conform to a normal distribution, thus the resultant statistical tests were non-parametric in nature.

- Correct. We have amended the statement to:

“Statistical testing was carried out using the IBM SPSS Statistics 26 software package. Shapiro-Wilks test for normality indicated $\delta^{13}\text{C}$ and $\delta^{15}\text{N}$ data did not conform to a normal distribution ($P<0.01$), thus the resultant statistical tests were non-parametric in nature.” (Lines 121-123)

Line 187 – Please report the full results of pairwise comparisons and not just p values.

- Amended for this and all other statistical tests.

Lines 192 – 202 – This is all interpretation and belongs in the discussion, not the results.

- To avoid repetition and make for a more concise publication, our manuscript presents a combined Results and Discussion section.

Lines 205 – 209 – Again this is discussion and not results.

- See above.

Lines 219 – 220 – You cannot just remove data without a rationale and statistically grounded justification.

- We now no longer present the test with the omission of sample SH12. The section reads:

“A significant positive correlation was observed between $\Delta^{15}\text{N}_{(\text{skin-bone})}$ values and age ($r=0.50$, $n=25$, $P=0.010$), however this was heavily influenced by the oldest individual (SH12, five years old, $\Delta^{15}\text{N}_{(\text{skin-bone})} = +2.9\text{‰}$), which had been in poor health prior to death (see Section 3.5).” (Lines 188-190)

Line 228 – I don’t understand what you mean by ‘turnover discrimination’. Consider rephrasing.

- For clarity, we have rephrased this sentence to: “This suggests that the variation between these tissues is present regardless of turnover rate...” (Lines 195-6)

Lines 234 – 236 – You need to report $\delta^{13}\text{C}$ and $\delta^{15}\text{N}$ values to guide the reader through the directionality of the seasonal effects. Also which statistical analyses indicated differences across season?

- These comments refer to analysis around seasonal variation in bone. Following comments by Reviewer 1 about the influence of reference 19, this section has been removed.

Line 253 – 254 – Results of statistical analyses are not reported.

- This section now reads: “The skins of two stillborn lambs (SH10, SH11) and a two-day old lamb (SH09) all showed a depletion in ^{13}C (mean: -0.8‰) and enrichment in ^{15}N (mean: $+0.8\text{‰}$) consistent with the wider population.”

Appendix B

Author's Response to Reviewers Comments

Reviewer: 1

The authors have done a thorough and admirable job addressing my comments.

My only input at this stage is to correct the grammar for the statement below, and to consider two relevant citations for it:

"We instead postulate the consistent routing of carbon and nitrogen that differs between skin and bone, with potentially on-site synthesis of non-essential amino acids occurring carbon and nitrogen that have been sourced via different biochemical pathways."

Citations:

Jim et al. 2006. Quantifying dietary macronutrient sources of carbon for bone collagen biosynthesis using natural abundance stable carbon isotope analysis. *British J Nutrition* 95:1055-1062

Newsome et al. 2014. Amino acid $\delta^{13}\text{C}$ analysis shows flexibility in the routing of dietary protein and lipids to the tissue of an omnivore. *Integr and Comp Biol* 54:890-902.

- We thank the reviewer for their comments on the manuscript. In light of their suggestion, the sentence has been amended to:

"We instead postulate a consistent difference in the routing of dietary protein and lipids between skin and bone, with potentially on-site synthesis of non-essential amino acids using carbon and nitrogen that have been sourced via different biochemical pathways [59,60]."

We thank the reviewer for drawing our attention to these two publications. These have been cited in-text and added to the bibliography – [59] Jim et al. 2006, [60] Newsome et al. 2014.